# The G-OBIM tongue model: An accurate open-source biomechanical model of a male human tongue

Maxime Calka [ID][1,2,3*], Pierre Badin [ID][2], Mohammad Ali Nazari[4], Michel Rochette[3], Pascal Perrier [ID][2], Yohan Payan [ID][1]

**1** Université Grenoble Alpes, CNRS, Grenoble INP, TIMC, Grenoble, France, **2** Université Grenoble Alpes, CNRS, Grenoble INP, GIPSA-Lab, Grenoble, France, **3** ANSYS, Villeurbanne, France, **4** Department of Mechanical Engineering, Faculty of Engineering, University of Tehran, Tehran, Iran

\* maxime.calka@univ-grenoble-alpes.fr

**Data availability statement:** All the data and code we used in our work are available on a

## Abstract

*Background and objectives:* We present a new Finite Element (FE) tongue model that was designed to precisely account for 3D tongue shapes produced during isolated French speech sounds by a male individual (RS). Such a high degree of realism will enable scientists to precisely and quantitatively assess, in a speaker-specific manner, hypotheses about speech motor control and the impact of tongue anatomy, muscle arrangements, and tongue dynamics in this context.

*Methods:* The shape and topology of the FE model were generated from 3D high resolution orofacial MR images of RS having his tongue in "neutral" posture. Mesh density was determined with convergence and mesh quality analyses. In a first step, muscle anatomy in the tongue was determined based on existing literature, and, in a second step, it was refined and evaluated by comparing actual and simulated 3D tongue shapes for various French speech sounds.

*Results:* Results are twofold. Firstly, a functional organization of the Genioglossus muscle into 4 parts is proposed which, on the one hand, is compatible with anatomical observations of the human tongue, and, on the other hand, goes beyond this anatomical account to faithfully reproduce the 3D tongue shape observed for RS in vowel /i/. Secondly, the realism of this implementation is demonstrated by the good match obtained for other isolated French sounds between actual and simulated tongue shapes. These simulations also inform us about the recruitment of tongue muscles for the main French speech sounds. Recruitment patterns are consistent with findings from the literature including both EMG measurements and model-based simulations.

*Conclusion:* The new model is made freely available, along with the data. Combined with mathematical tools that transform the tongue model cloning RS tongue into other models that account for the morphology of various individuals, the model can be a powerful tool to investigate healthy and pathological speech from various perspectives.

cloud provided by the CNRS (Janus sDrive; https://sdrive.cnrs.fr/s/3reaKAmg4jii7Q7). Due to the ethical agreement under which they were recorded, given their medical nature, MRI data will only be accessible upon request, and cannot be publicly shared. To get access to these data, please contact the direction of Gipsa-lab: direction@gipsa-lab.fr.

**Funding:** This work was supported by the French "Agence Nationale de la Recherche" within: (1) The "Multidisciplinary Institute in Artificial Intelligence" at Grenoble Alpes Université (ANR-19-P3IA-0003 - MIAI@Grenoble Alpes – Director: Eric Gaussier, to PP, who hold the Chair "Bayesian Cognition and Machine Learning for Speech Communication"). (2) The "e-SwallHome – Swallowing and Respiration: Modeling and e-Health at Home" project ("Technologies pour la Santé" program, ANR-13-TECS-0011, PI: Jacques Demongeot, to PB, PP, YP). (3) The "Investissements d'Avenir" program (ANR-11-LABX-0004 - Labex CAMI – PI: Jocelyne TROCCAZ, to YP)). MC's salary was supported by the French "Agence Nationale de la Recherche Technologique" (ANRT) and the company "ANSYS France", in the context of a CIFRE contract. The "Agence Nationale de la Recherche" and the "Agence Nationale de la Recherche Technologique" (ANRT) played no role in study design, data collection and analysis, decision to publish, or preparation of the manuscript. ANSYS provided free access to the ANSYS FE package and contributed its expertise on the Finite Element Method and the use of the FE package.

**Competing interests:** The authors have declared that no competing interests exist.

## Author summary

Over the past decades, biomechanical Finite Element models of the tongue have evolved to better understand speech production. This study presents a new tongue model, specifically designed to accurately replicate 3D tongue shapes of a reference French speaker during sustained speech sounds. This model is part of the *Grenoble Orofacial Biomechanical Model* (G-OBiM), which aims at modeling all the organs involved in speech production and swallowing. Using high-quality MRI data from the reference subject, the design of the *G-OBIM tongue model* drew on the various models developed over the last 20 years within our research group to propose refined anatomical and functional representations of the tongue muscles, validated through simulations. In particular, a four-part functional organization of the Genioglossus muscle is proposed to better capture the dynamics of the tongue. The model makes it possible to simulate, with a degree of precision unprecedented in the literature, 3D tongue shapes achieved by the reference speaker during the production of various French speech sounds. By offering information on associated muscle activation patterns, it serves as a valuable tool for speech research and is freely available to the scientific community to explore healthy and pathological speech production.

## 1. Introduction

Since the early 1970s, several biomechanical models of the human tongue embedded in the vocal tract have been developed. These models were used mainly to investigate and characterize the influence of oral biomechanics and the anatomical organization of the tongue on the following:

1. the role of tongue muscles in tongue shaping [1,2].
2. the characteristics of tongue shapes associated with the most frequent vowels [3–5].
3. the spatio-temporal properties of tongue movements in speech production [5–7].
4. general aspects of speech motor control [8–11].
5. tongue muscle activations in vowel production [9,12,13].
6. motor control of swallowing [14].
7. the consequences of tongue surgery on speech production [15,16].
8. the role of tongue control in sleep apnea [17].

All of these contributions have been important in improving our general knowledge of the crucial role of the tongue in fundamental orofacial biological functions that are central to human quality of life: breathing, swallowing, and speaking. They have been obtained over the years thanks to an increasing level of complexity in the account of tongue tissue physical properties, which has been made possible by tremendous improvements in computing capacities and numerical methods: from a discrete spring-based [10,18] to a continuum mechanics representation [2,6] of the tongue body; from a linear elastic [4] to a non-linear hyperelastic [1,2] account of the mechanical properties of the tongue; from a small set of large elements [3] to a high number of small elements [19] to design the 3D Finite Element (FE) mesh that represents tongue geometry; from a functional modeling of muscle activations with external forces applied onto specific nodes of the 3D FE mesh [3,6,9] to a continuous active stress generated by elements that represent muscles in the 3D mesh [11,17,19].

In the current work, we propose a new three-dimensional model of the tongue embedded in the vocal tract. This model is part of the *Grenoble Orofacial Biomechanical Model* (G-OBiM), which aims at modeling all the organs involved in speech production and swallowing. It will be called *G-OBIM tongue model* henceforth. Its foundations lie in the successive work carried out in our group by Gerard, Buchaillard, Hermant and colleagues [1,9,19]. As in these previous models, the geometry and topology of the model have been determined from anatomical data of the tongue recorded on a French male individual (born 1955) in a posture considered neutral, and the mechanical behavior of the soft tissues of the tongue is modeled by a non-linear constitutive law, in a large deformations framework. The novelty of our approach lies in the objective of developing a model that faithfully represents the tongue of this male individual in both morphological and anatomical terms. This will allow us to precisely study how he controls his tongue during speech by quantitatively comparing articulatory data collected on this individual with simulations of the same speech sequences performed using the model. It is important to note that, over the last three decades, this individual has agreed to serve as a subject in numerous experimental studies on speech articulation and acoustics. For this individual, we can therefore refer to numerous data sets, recorded during speech production by X-ray cineradiography, electromagnetic articulography, anatomical Magnetic Resonance Imaging (MRI) and video, along with the acoustic speech signal. This opens up a wide range of perspectives for quantitative evaluation, in a speaker-specific manner, of theoretical hypotheses and models of speech production and speech motor control. This individual can therefore be considered a true *reference subject* for the study of speech articulation and speech motor control, and we now call him "Reference Subject" (RS). With this new, highly precise model, we aim to go beyond the generic description of the fundamental mechanical properties of the tongue and their influence on general aspects of speech production and motor control, as proposed in the literature to date.

From a methodological point of view, our approach involved several stages. First, we used *high resolution* 3D MRI data, which provide a better account of RS orofacial anatomy. A tetrahedral FE mesh of the RS tongue was designed from the set of MRI images representing the tongue in neutral position. Secondly, to determine the accuracy of the mesh, we used a *convergence analysis* and searched for the number of nodes and elements that provided similar accuracy to the devices used to record articulatory data. Thirdly, we used a *more realistic 3D mechanical muscle model* [20]. Fourthly, building on the implementation proposed in our previous models [9,19] and using anatomical data from the literature (in particular [21], [22] and [23]) we *refined the anatomical implementation of tongue muscles* in the model. To do so, we used the set of MRI images representing RS tongue during the production of the French vowel /i/ pronounced in isolation and relied on an interactive approach between simulations with the model and observations of the MRI data. In particular we proposed a description of the Genioglossus in four functional units. Finally, the refined muscle implementation was *assessed for other French phonemes* pronounced in isolation by comparing the corresponding MRI data to simulations carried out with the model.

Thus, more than a continuation of previous modeling works carried out in our research group, the FE model presented in this article represents a new step in our approach, both in terms of modeling (topological and anatomical description especially for muscle like the Genioglossus muscle model) and quantitative evaluation, in the aim to simulate with a satisfying level of accuracy speech articulation in selected human subjects. Moreover, we propose to make this new version of the FE model available to the academic community (Moreover, this new version of the FE model is freely available to the academic community and the set of MRI images of RS' speech articulations is available upon request - see Data Availability) as well as the set of MR images data that have been collected on speech articulations of RS.

## 2. Results

### 2.1. Convergence analysis

In this analysis we considered the absolute errors (i.e. Euclidean distance in mm) in the maximal displacements of the four nodes of interest (in the apical, alveolar, velar and pharyngeal regions) predicted in the simulations with the different meshes as compared to the reference mesh, the one with the highest resolution. Results in the apical, alveolar, velar and pharyngeal parts of the tongue are given in Table 1. Complementary results such as the convergence curve or the relative errors are given in Fig A and Table A in S1 Text. Given the critical error threshold of 1 mm (see Sect 4.5) and in light of the values reported in Table 1, the mesh composed of 57034 nodes can be considered as the most appropriate, since the absolute errors are around 0.5 mm. Among the evaluated meshes, it provides the best compromise between simulation accuracy and computational cost.

Based on this specific study, we concluded that a mesh made of around 60000 nodes should be dense enough to ensure an accuracy of the simulations that enables relevant comparisons between the simulated node displacements over time and actual articulatory displacements measured on a human subject. Since the six tested meshes have been generated by starting from a very coarse one (obtained with a strong undersampling of the tongue surface) and by dividing then the elements of this initial mesh, the topologies of these meshes slightly differed from the actual topology of the surface of the tongue of RS. Hence, instead of considering the mesh with 57034 nodes, it was decided to generate a new mesh with around 60000 elements starting from the initial tongue mesh topology (see Fig 16) and remeshing the volume thus defined using a renowned and highly reliable commercial software, *Hypermesh*. The final mesh, composed of 61117 nodes and 41600 quadratic tetrahedral elements, is presented below in Fig 1. An element quality analysis on the final mesh has been carried out, which validates this design. It is presented in Tables B, C and D in S1 Text.

**2.1.1. Full model of tongue and oral cavity.** The full model of the tongue and the oral cavity is shown in Fig 1. The numbers of elements and nodes of each of the represented anatomical structures are listed in Table 2, together with the type of elements and contacts. In the tongue mesh, the quadratic tetrahedral elements have an average size of 1.79 mm with a standard deviation of 0.32 mm. The maximum and minimum element sizes are 3.25 mm and 0.84 mm respectively.

As concerns the other anatomical structures, their element sizes have been chosen to be similar, in order to facilitate the implementation of mechanical contacts with the tongue mesh.

**Table 1**. Absolute error (in mm) obtained with the different meshes for the considered four node positions in the apical, alveolar, velar and pharyngeal regions, as compared to the reference simulation with the most accurate mesh (the one with 3412064 nodes).

| Number of nodes | Absolute error (in mm) | | | |
|---|---|---|---|---|
| | Apical | Alveolar | Velar | Pharyngeal |
| 7795 | 0.913 | 1.247 | 1.082 | 1.035 |
| 57034 | 0.368 | 0.522 | 0.450 | 0.432 |
| 187051 | 0.211 | 0.309 | 0.265 | 0.254 |
| 436221 | 0.121 | 0.179 | 0.153 | 0.146 |
| 1450427 | 0.038 | 0.053 | 0.046 | 0.044 |
| 3412064 | Reference | | | |

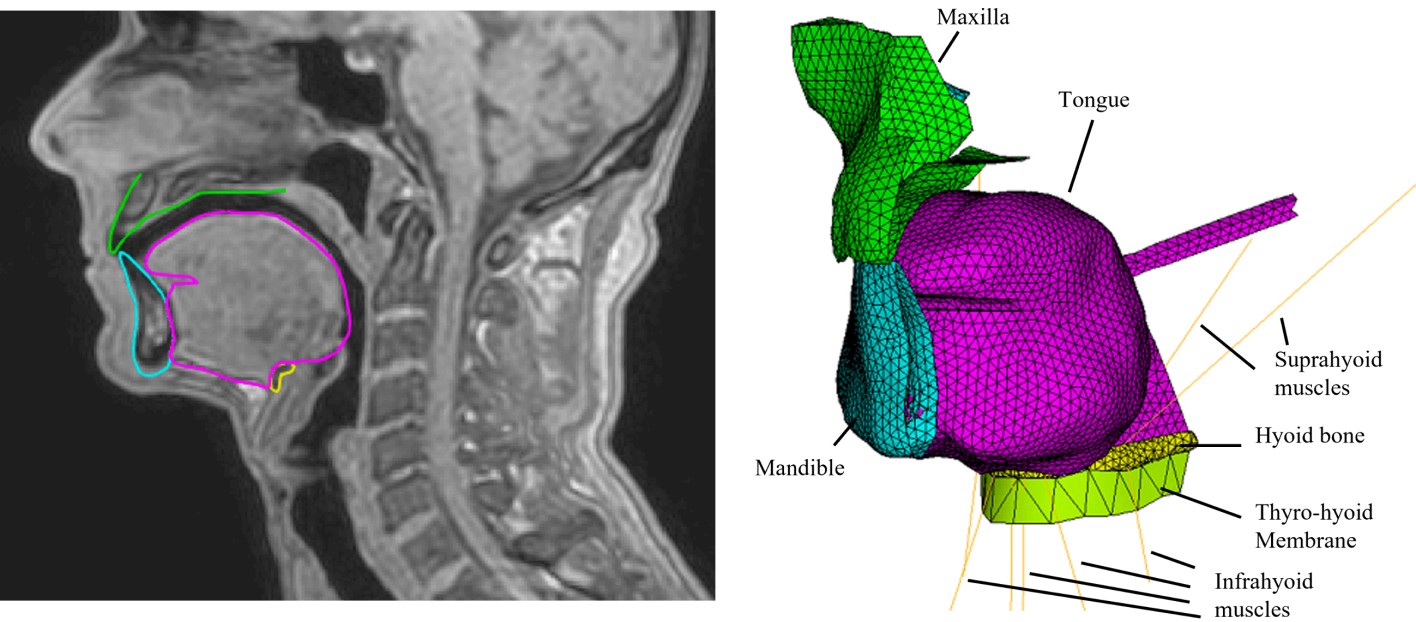

**Fig 1. 3D FE model of the oral cavity.** Left: the FE model superimposed on the MRI data of RS in the mid-sagittal plane (Front is on the left). Right : Oblique anterior view of the model of the oral cavity with the tongue in rest position (magenta), the mandible (cyan), the hyoid bone (yellow), the maxilla (green), the hyoid muscles represented with lines (orange), and the thyro-hyoid membrane (light green).

**Table 2**. **Numbers of nodes and elements, and types of elements for each of the meshed anatomical structures implemented in the** *ANSYS MAPDL* **environment.** The types of contact between these structures are also listed.

| | Number of nodes | Number of elements | Element type |
|---|---|---|---|
| Tongue | 61117 | 41600 | quadratic tetrahedron (SOLID187) |
| *standard contact with palate* | 17273 | 14282 | quadratic triangle (CONTA174) |
| *standard contact with mandible* | 1916 | 2204 | quadratic triangle (CONTA174) |
| *bonded contact with mandible* | 399 | 288 | quadratic triangle (CONTA174) |
| *bonded contact with hyoid bone* | 425 | 176 | quadratic triangle (CONTA174) |
| Hyoid bone | 545 | 1086 | linear triangle (SHELL181) |
| *including target elements for contact* | 294 | 474 | linear triangle (TARGE170) |
| Mandible | 3540 | 7076 | linear triangle (SHELL181) |
| *standard contact with mandibular fossae* | 186 | 640 | linear triangle (CONTA173) |
| *including target elements (standard contact)* | 635 | 2288 | linear triangle (CONTA174) |
| *including target elements (bonded contact)* | 214 | 624 | linear triangle (TARGE170) |
| Hard palate | 1826 | 3534 | linear triangle (SHELL181) |
| *including target elements for contact* | 619 | 2296 | linear triangle (TARGE170) |
| Mandibular fossae | 588 | 520 | linear quadrangle (TARGE170) |
| Infra- and supra- hyoid muscles | 22 | 12 | 3D beam (LINK180) |
| Thyro-hyoid membrane | 26 | 24 | linear triangle (SHELL181) |

## 2.2. Muscle implementation

### 2.2.1. Functional partitioning of the Genioglossus.

To determine the functional units of the Genioglossus we simulated the production of the French /i/ by RS using an empirical trial and error adjustment of tongue muscle activations (performing an inverse analysis would be too time-consuming, given the duration of a simulation, in this case several hours), once the mandibule and hyoid bone positions have been adapted to the ones observed in the MRI data recorded for this vowel (see below Sect 2.3 for details about the method in this regard). As a

starting point in this process, we activated the tongue muscles following the experimental and modeling results published in the literature [3,18,24] and the modeling findings obtained with former versions of our tongue model [9].

Thus, the anterior (GGa) and posterior (GGp and GGh) parts of the Genioglossus and the Geniohyoid muscles were activated. The Geniohyoid contributes to enlarge the pharyngeal cavity. In the upper right panel (v) of Fig 2 the MR image reveals the existence of two zones of compression in the tongue: one in the anterior part, at the level of the grooving frequently observed during the pronunciation of /i/ and nicely accounted for in Buchaillard et al. [9] by the activation of the GGa, and the other one in the posterior part of the tongue. In line with

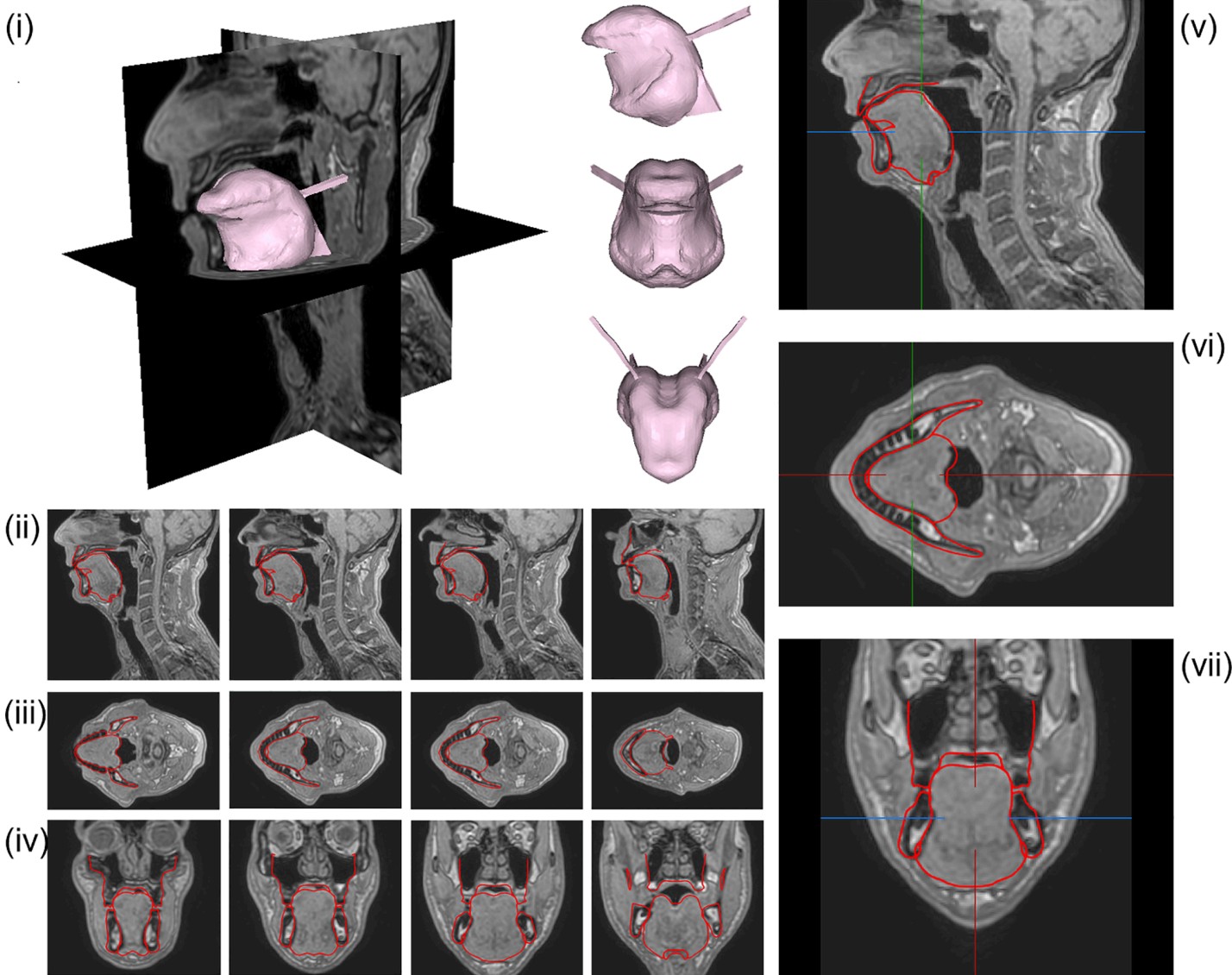

**Fig 2. Vowel /i/ generated with the model using the Genioglossus partitioning shown in Fig 3.** Contours of the *G-OBIM tongue model* (in red) in the steady-state postures associated with the articulation of these sounds superimposed to various MR slices: (i) FE model of the tongue in the steady-state posture; (ii) sagittal views (left to right: mid-sagittal to left lateral); (iii) axial views (left to right: inferior to superior); (iv) coronal views (left to right: posterior to anterior); (v) enlargement of the tongue region in the mid-sagittal, (vi) axial and (vii) coronal views.

Buchaillard et al. [9], we reproduced the compression in the anterior part by activating the GGa, and we adapted the boundaries of this functional unit of the Genioglossus to faithfully reproduce the 3D tongue surface extracted from the MRI data in this region. Similarly, we reproduced the posterior compression by assuming that it is mainly due to the activation of the GGp and GGh, and we adapted the boundaries of the GGp to faithfully reproduce the 3D tongue surface in this region.

The resulting partitioning of the Genioglossus in the model is shown in Fig 3. The 3D simulated tongue configuration obtained for vowel /i/ with this partitioning is shown in the upper left panel (i) of Fig 2. In the upper right panel (v), the sagittal contour of the tongue configuration is superimposed on the sagittal view of the vocal tract extracted from the MRI data.

**2.2.2. Other muscles implementation.** Figs 4 to 11 show the implementation of the other muscles of the tongue and the ones of the mouth floor. This implementation was defined on the basis of the literature and enabled us to find the main tongue movements linked to their activations, as shown in Figs F to Q in S1 Text.

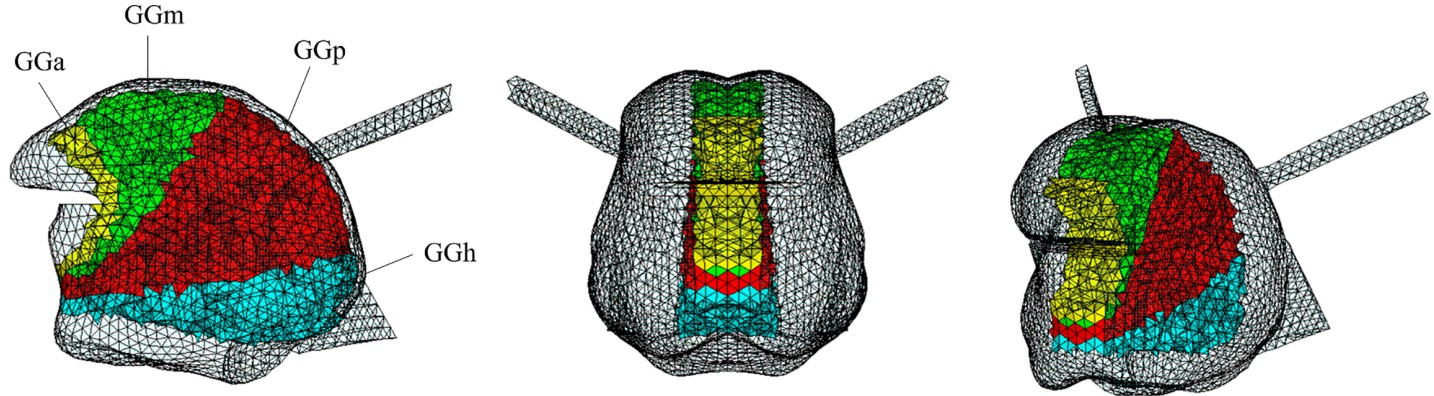

**Fig 3. Proposed functional partitioning of the Genioglossus resulting from the reproduction with the model of the articulation of vowel /i/ by RS.** GGa: yellow; GGm: green; GGp: red; GGh: cyan. Left: sagittal view; center: coronal; right: cavalier perspective view.

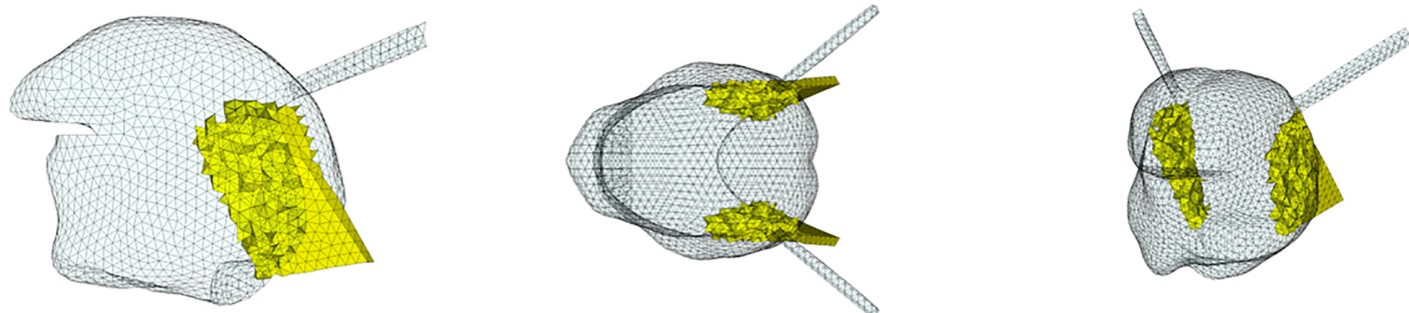

**Fig 4. Topological description of tongue muscles: Hyoglossus (HG).** Left: sagittal view; center: coronal or axial view; right: cavalier perspective view.

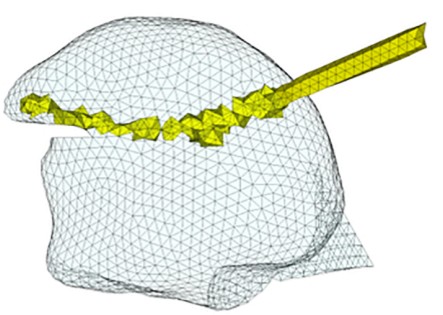
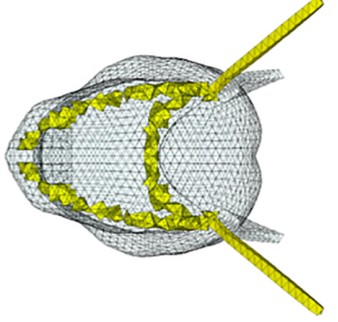
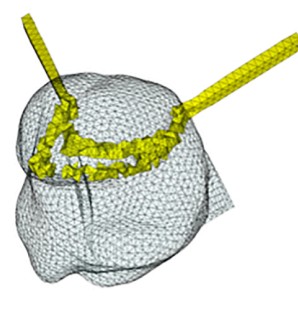

**Fig 5. Topological description of tongue muscles: Styloglossus (SG).** See Fig 4 for details.

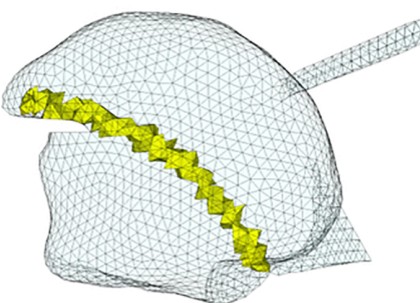
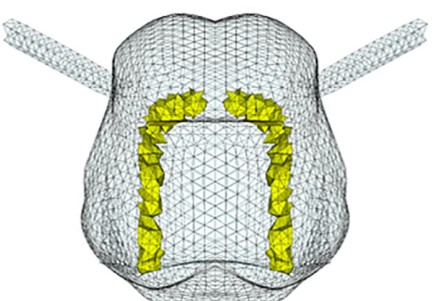
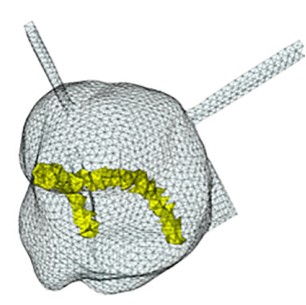

**Fig 6. Topological description of tongue muscles: Inferior Longitudinalis (IL).** See Fig 4 for details.

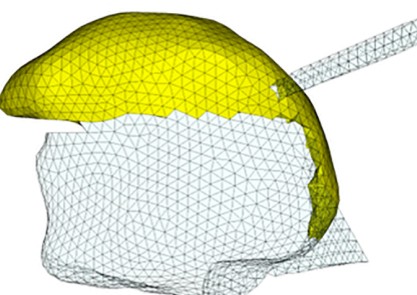
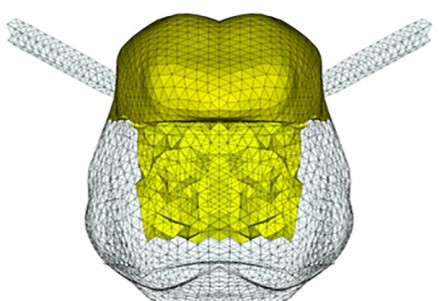
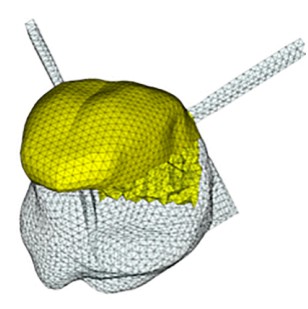

**Fig 7. Topological description of tongue muscles: Superior Longitudinalis (SL).** See Fig 4 for details.

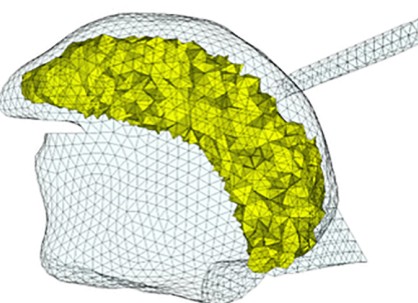
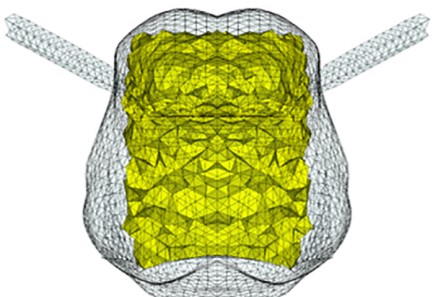
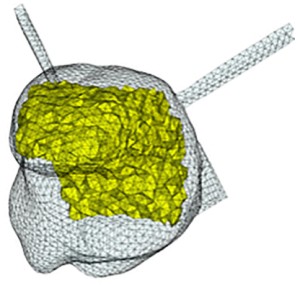

**Fig 8. Topological description of tongue muscles: Transversalis (Trans).** See Fig 4 for details.

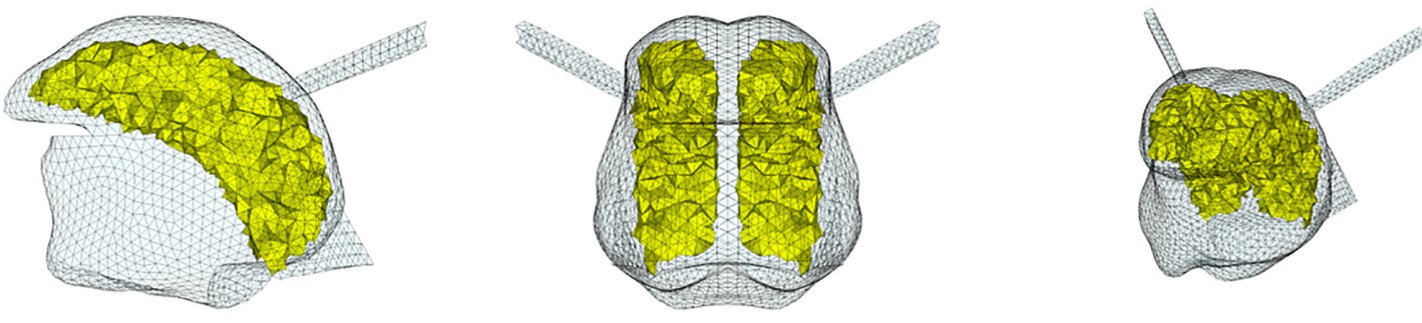

**Fig 9. Topological description of tongue muscles: Verticalis (Vert).** See Fig 4 for details.

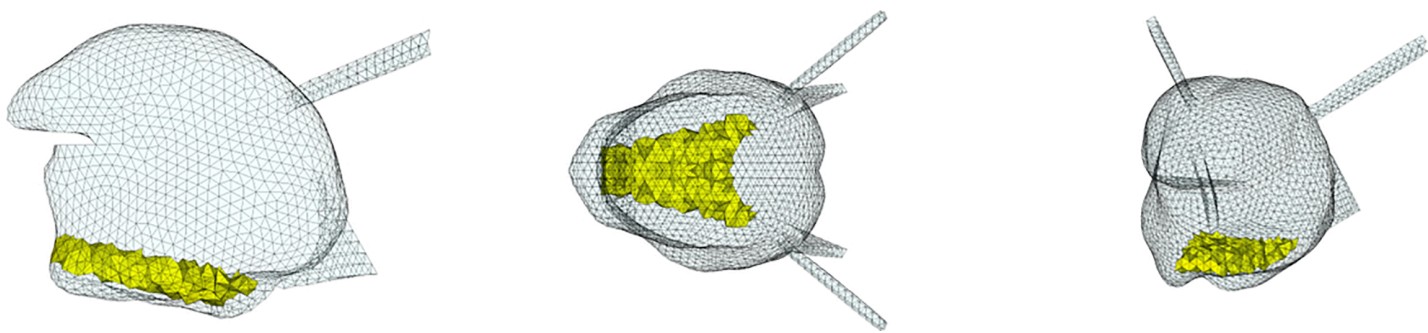

**Fig 10. Topological description of muscles acting on the tongue in the mouth floor: Geniohyoid (GH).** Left: sagittal view; center: coronal or axial view; right: cavalier perspective view.

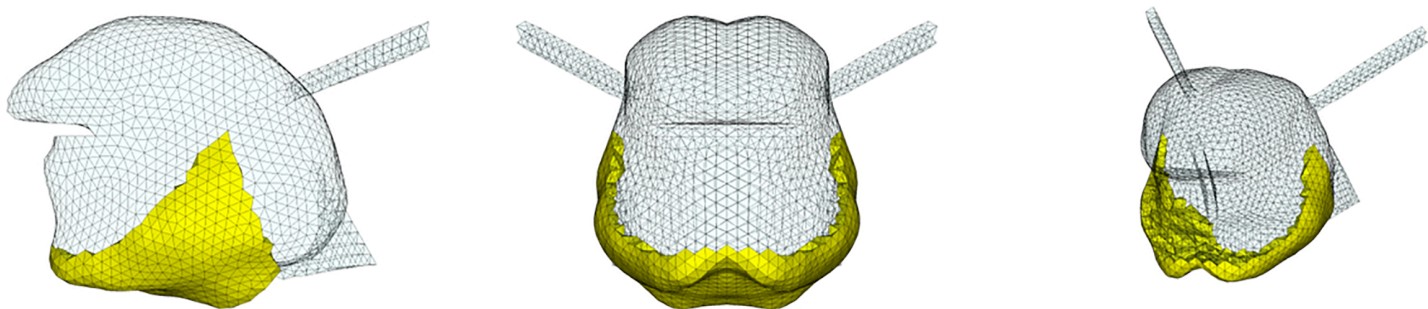

**Fig 11. Topological description of muscles acting on the tongue in the mouth floor: Mylohyoid (MH).** See Fig 10 for details.

## 2.3. Evaluation of the model: Simulation of /a/, /u/, /t/ and /k/

In this section, we had two main objectives: the validation of the proposed functional partitioning of the Genioglossus and an assessment of the overall quality of our model by quantitatively comparing its simulations with precise articulatory data.

We selected four French phonemes: /a/, /u/, /k/, and /t/, in addition to the phoneme /i/. These phonemes were chosen for their specific articulatory characteristics. Indeed, /i/, /a/, and /u/ are the extreme cardinal vowels found across all the world's languages. As for /k/ and /t/, they are both stop consonants: the former is a posterior (velar) consonant, and the latter an anterior (alveo-dental) one.

To simulate the French /a/, /u/, /k/ and /t/, as produced by RS, a preliminary study of the position of the mandible and hyoid bone was carried out (Table 3). The idea was to start from positions of the hyoid bone (imposed displacement in the vertical direction associated with Geniohyoid and Genioglossus activations for the horizontal displacement) and the mandible (imposed displacement of the mandible 3D mesh) similar to the positions observed in the MR image. Once a satisfactory position of the mandible was reached, tongue muscles were activated to produce each elementary articulation. Table 4 shows the corresponding active stresses exerted by each muscle.

Figs 12 to 14 (whose organization is inspired by [25]) show the results of simulations aiming to reproduce the extreme cardinal vowels /a/ and /u/ and the unvoiced stop consonants /k/ and /t/, as they were articulated by RS. These results were obtained empirically with *ad hoc* trial and error adjustments of tongue muscles' activations, taking as a starting point the muscle patterns proposed under similar conditions by [9]. Methods such as inversion are not feasible at this stage because they would require many simulations to learn the relationships between muscle activations and mesh node positions, which would be prohibitive in terms of computation time.

**Table 3**. **Mandibular opening (in mm) resulting from imposed displacements to match the mandible position in the MRI data.** Displacement of the hyoid bone (in mm) resulting from the activations of the Geniohyoid and Mylohyoid muscles and from imposed displacements in vertical directions. Displacement along X: front-back; displacement along Y: up-down.

| Mandibular opening (in mm) | | Displacement in X direction of the hyoid bone (in mm) | Displacement in Y direction of the hyoid bone (in mm) |
|---|---|---|---|
| /a/ | 11.0 | 0.8 (front) | 0.9 (bottom) |
| /u/ | 7.0 | 6.0 (front) | 7.0 (bottom) |
| /i/ | 3.0 | 5.8 (front) | 5.6 (bottom) |
| /t/ | 1.0 | 1.1 (front) | 3.3 (top) |
| /k/ | 5.0 | 2.7 (front) | 5.2 (top) |

**Table 4**. **Active stresses (in Pa) exerted by each muscle in the model to reproduce the articulation of /a/, /u/, /i/, /t/ and /k/ by RS.**

| | GGa | GGm | GGp | GGh | SG | HG | Vert | Trans | IL | SL | GH | MH |
|---|---|---|---|---|---|---|---|---|---|---|---|---|
| /a/ | 8000 | 8000 | 0 | 2000 | 20000 | 3000 | 0 | 0 | 5000 | 0 | 0 | 0 |
| /u/ | 0 | 0 | 0 | 35000 | 90000 | 0 | 0 | 0 | 0 | 2000 | 10000 | 0 |
| /i/ | 3500 | 0 | 4000 | 10000 | 0 | 0 | 0 | 0 | 0 | 0 | 6000 | 0 |
| /t/ | 15000 | 10000 | 2000 | 5000 | 0 | 0 | 500 | 0 | 0 | 1300 | 0 | 5000 |
| /k/ | 0 | 0 | 0 | 6000 | 70000 | 0 | 0 | 0 | 0 | 0 | 2500 | 10000 |

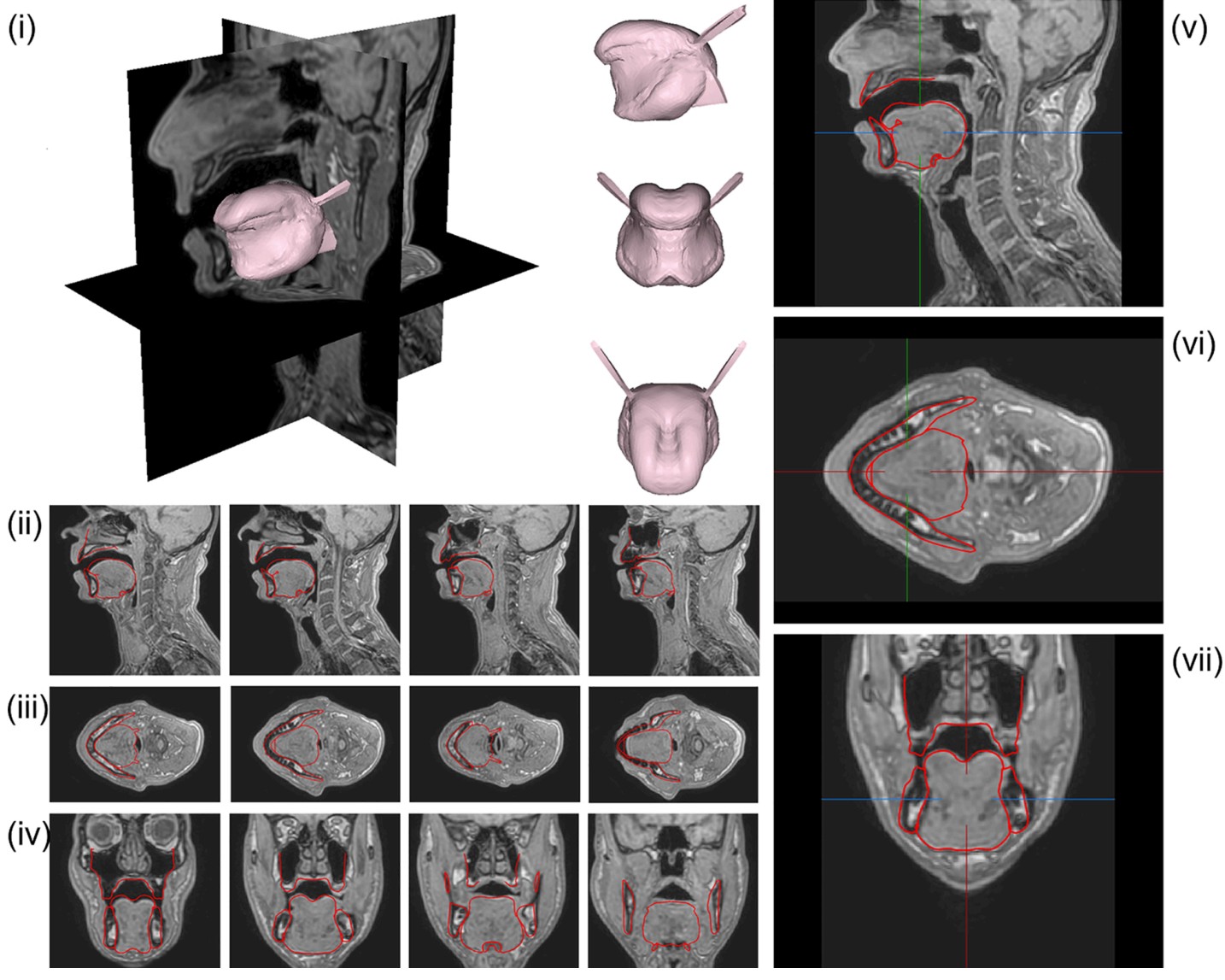

**Fig 12. Vowel /a/.** Contours of the *G-OBIM tongue model* (in red) in the steady-state postures associated with the articulation of these sounds superimposed to various MR slices: (i) FE model of the tongue in the steady-state posture; (ii) Sagittal views (left to right: mid-sagittal to left lateral); (iii) Axial views (left to right: inferior to superior); (iv) Coronal views (left to right: posterior to anterior); (v) Enlargement of the tongue region in the mid-sagittal plane, (vi) axial and (vii) coronal views.

**Vowel /a/**

To produce the vowel /a/, the anterior (GGa), medium (GGm), horizontal (GGh) Genioglossus, the Styloglossus, Inferior Longitudinalis and Hyoglossus muscles are activated. Their role can be interpreted in the light of the results presented in Fig F to Q in S1 Text. The Hyoglossus is usually considered as a major muscle in the production of /a/ [9,24] because it moves the tongue body downwards and backwards, which opens the vocal tract

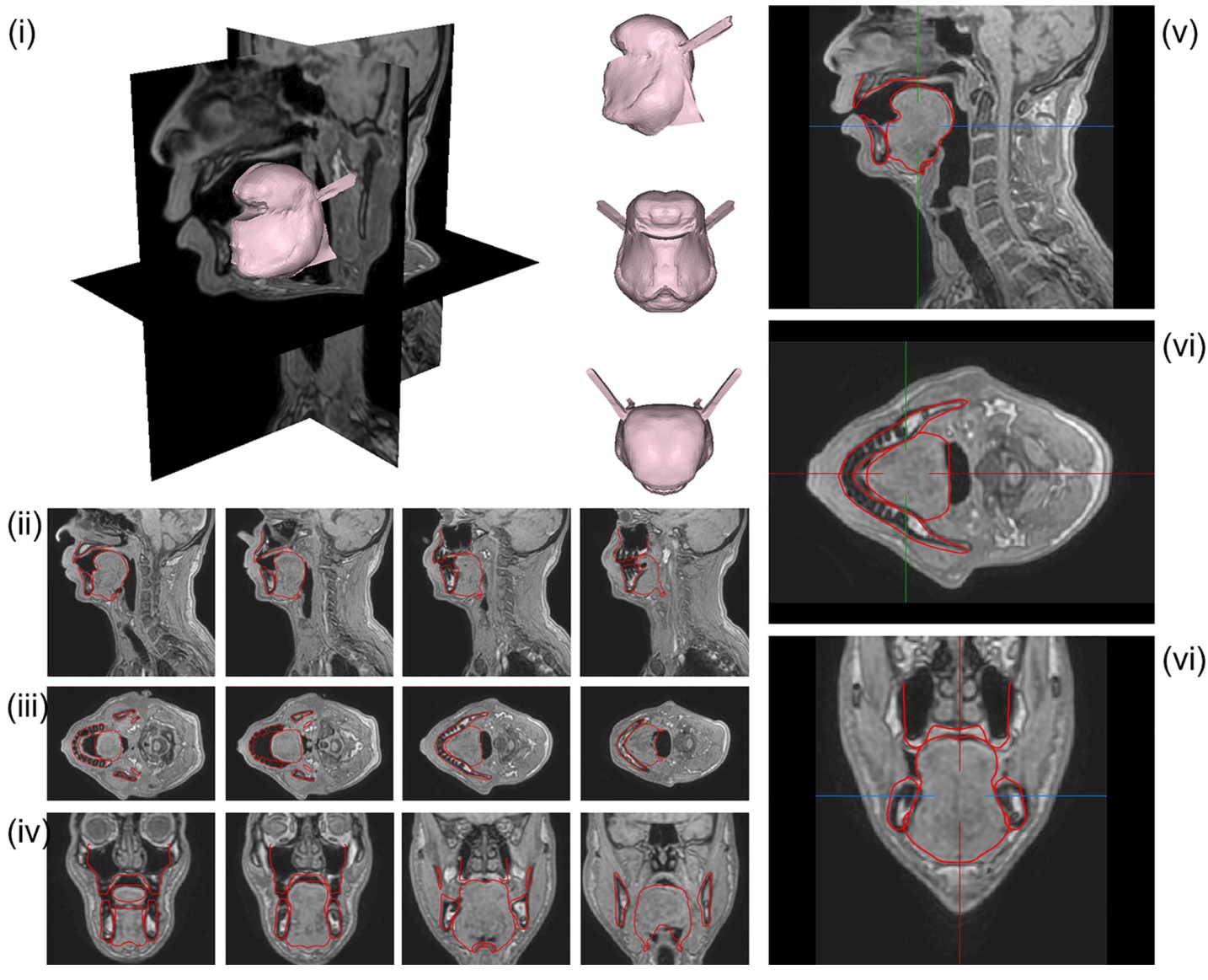

**Fig 13. Vowel /u/.** See Fig 12 for details.

in its front part and narrows it in the pharyngeal region, which corresponds to typical characteristics of this vowel [26]. In the articulation produced by RS, the activation of the Styloglossus also contributes to the displacement of the tongue body to the back. The activation of the GGh generates an elevation of the back part of the tongue dorsum. This elevation is counteracted by the activation of the GGm, which has been proposed in the literature [10] to form with the Styloglossus an antagonist muscle pair acting in synergy in the tongue dorsum region. Finally, the cross-sectional area of the front cavity of the vocal tract is enlarged thanks to the lowering of the tongue tip that is generated by the combined action of the GGa, the anterior fibers of Styloglossus, and the Inferior Longitudinalis that also move the tongue tip back toward the tongue body. In general, the approximation of the tongue shape provided by the model with the proposed activations is pretty accurate (Fig 12), especially in the

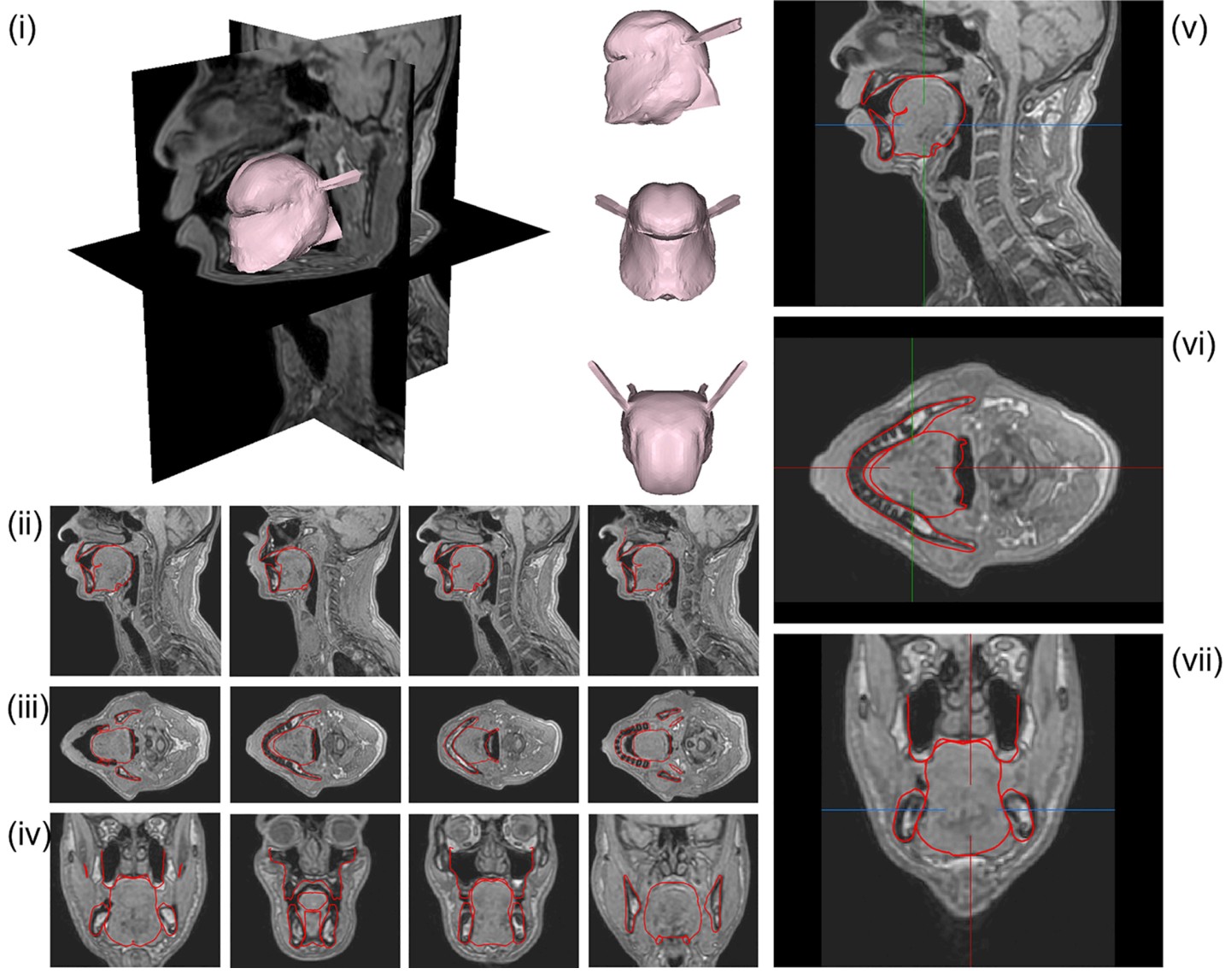

**Fig 14. Stop Consonant /k/.** See Fig 12 for details.

so-called *mobile tongue*, which goes from the tongue tip to the upper part of the pharyngeal region.

### Vowel /u/

To produce the vowel /u/, the horizontal Genioglossus (GGh), the Styloglossus, the Superior Longitudinalis and the Geniohyoid muscles are activated. The Styloglossus is the most important muscle in this vowel because it elevates the tongue dorsum in the velar region of the oral cavity, and lowers and retracts the tongue tip (see Fig F to Q in S1 Text), which are typical characteristics of the articulation of this vowel [26]. The Styloglossus also moves the tongue body backwards. The GGh contributes to the elevation of the tongue dorsum, in combination with the action of the Styloglossus, and induces also an elevation of the tongue blade. Interestingly, it does not seem useful to activate any other part of the Genioglossus than the

GGh. This tends to justify the separation in the back part of the Genioglossus of the GGp and the GGh. The activation of the GGh is required to generate the required compression of the tongue in its posterior part, which enlarges the cross-sectional area of the pharyngeal cavity. The activation of the Superior Longitudinalis partly limits the retraction of the tongue tip induced by the Styloglossus and enables a gradual transition from the constriction in the velar region to the front cavity of the vocal tract. As for vowel /i/, the Geniohyoid contribute to the enlargement of the pharyngeal cavity, by moving foward the hyoid bone and the tongue root. It should be noted that a lowering of hyoid bone (Table 3) is also required to lower the tongue root, in order to facilitate the compression of the lowest part of the tongue in the pharynx, and increase the cross-sectional area of the pharyngeal cavity. Here again, the approximation of the tongue shape provided by the model is pretty accurate (Fig 13).

### Stop consonant /k/

The muscles activated during the production of consonant /k/ are similar to those of the vowel /u/. This is consistent with the articulatory characteristics of these speech sounds, which are both associated with a narrowing of the vocal tract in the velar region [26]. Here again, the specific role of the horizontal part of the Genioglossus (GGh) should be emphasized, whose activation makes possible the tongue compression in the lowest part of the pharynx. The crucial difference is that, compared to vowel /u/, the stop consonant /k/ requires a full sealing of the vocal tract in the velar region, and then a stronger elevation of the tongue dorsum. This elevation is achieved thanks to the combination of a higher position of the hyoid bone and of the activation of the Mylohyoid muscle that stiffens the mouth floor and facilitates the volume expansion of the tongue upwards. Note also that, as compared to vowel /u/, stop consonant /k/ has a lower position of the tongue tip, which is consistent with the fact that the Superior Longitudinalis is not activated. Here again, the modeled tongue shape of stop consonant /k/ corresponds satisfactorily to the tongue shape in the *mobile tongue*, as described in the MRI data (Fig 14).

### Stop consonant /t/

To produce the stop consonant /t/, the four units of the Genioglossus, the Verticalis, the Superior Longitudinalis and the Mylohyoid are activated. All these activations are consistent with the crucial articulatory characteristics of this consonant, which requires a front and high position of the tongue, with a complete sealing in the alveolar region of the vocal tract, close to the upper incisors [26]. The specificity of this consonant is a contact between the tongue and the teeth of the maxilla, requiring the activation of the Superior Longitudinalis muscle. The activations of the posterior (GGp) and horizontal Genioglossus (GGh) are required, because they generate a forward displacement of the tongue body and an elevation of the tongue dorsum and the tongue blade. The activation of the Mylohoid also contributes to the forward movement of the tongue. Building up on the thus obtained anterior positioning of the tongue, the activations of the anterior (GGa) and medium Genioglossus (GGm) combined with the activation of the Verticalis are important, because they flatten and lower the tongue blade and the tongue dorsum, limiting the elevation induced by the activations of the GGp and the GGh. The elevation of the hyoid bone and the tongue root, and the activation of the Mylohyoid, which stiffens the mouth floor, contribute to the elevation of the tongue body (Fig 15).

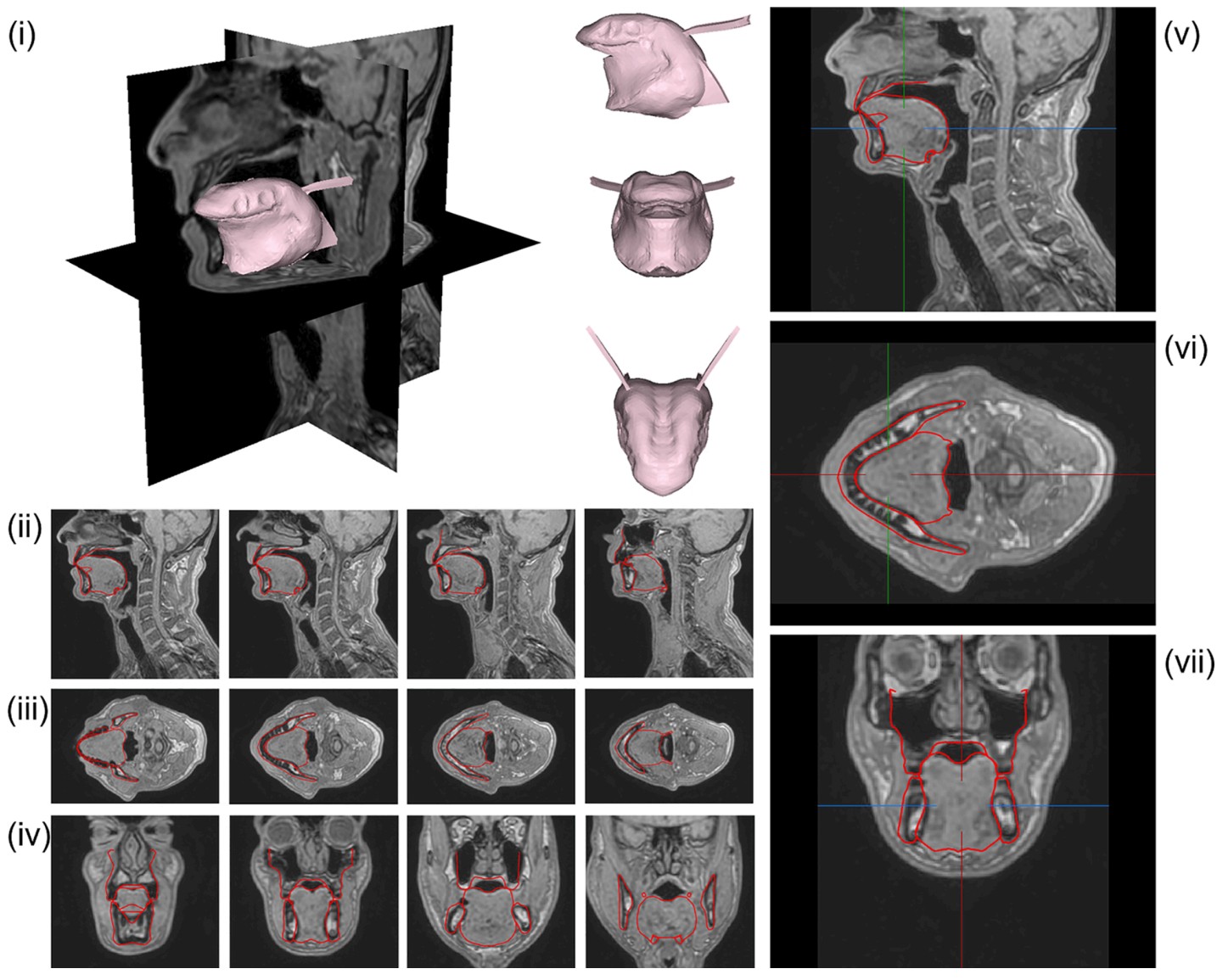

**Fig 15. Stop Consonant /t/.** See Fig 12 for details.

## 3. Discussion

As compared to our former biomechanical tongue models, the model presented in this paper includes a number of improvements that enables us to simulate accurately the articulation of speech sounds in our Reference Subject (RS). Some of them provide a better description of the morphology of the tongue, in particular in the apical region, with a more realistic description of the sub-apical cavity and a more independent mobility of the tongue tip. The convergence analysis has enabled us to determine a density of the FE mesh that ensures an accuracy of the transient FE simulations compatible with the accuracy of the measures of tongue displacements using electromagnetic articulography.

## 3.1. Implementation of the tongue muscles

In our quest for a model able to faithfully reproduce the tongue shapes produced by RS during the articulation of crucial French vowels and consonants, we were able to provide a significant contribution to a better understanding of the functional partitioning of the Genioglossus, the largest muscle of the tongue, which role is crucial for tongue mobility. Relying both on anatomical observations ([27], [22]), and electromyographic data [23], we decided to propose a decomposition of this muscle into 4 functional units: the Genioglossus horizontal (GGh), an anatomical region bounded by the mouth floor below and the short quasi-horizontal tendon attached to the superior genial tubercle above and 3 functional units (posterior, GGP; medium, GGm; anterior, GGa) in the fan-shaped part of the muscle, above the tendon, which reflect both EMG and histological data. The use of the accurate 3D description provided by the MRI data of the articulation of vowel /i/, in particular in the *mobile tongue*, has enabled us to provide insights into the boundaries of this partitioning. Two studies proposed to consider five independently controlled units in the Genioglossus ([28], [29]). Yet, none of them rely on any clear neuro-physiological evidence to support this partitioning. Both studies mentioned the EMG study of Miyawaki et al. [23] who used five electrodes to investigate the spatial distribution of activations in the Genioglossus. However, it is important to remember that this number of electrodes was arbitrarily chosen by Miyawaki et al., in order to demonstrate that "*the genioglossus muscle is, functionally, not "one muscle*""(page 101). Miyawaki et al. added: "*we do not know how many subdivisions there are in the genioglossus muscle, nor do we know in what manner the subdivisions are voluntarily controlled*" (page 101). In addition, in Miyawaki et al.'s data, the three anterior electrodes showed strong similarities in the time patterns of activations and so did also the two posterior electrodes. In his paper, Wrench mentioned a personal communication with Liancai Mu, who said "*The human GG appears to be composed of at least 4 NMCs* [NeuroMuscular Compartments], *including GGh and 3 GGo sub-compartments while considering muscle fibre arrangement, fibre type distribution (regionalization), and nerve supply patterns*" ([29], p. 3894). In fact, in order to justify a partitioning of the Genioglossus into more than four parts, Wrench [29] and Harandi et al. [28] looked for the partitioning that enabled the best fitting of tongue shapes observed during speech production. Wrench's work is interesting but its relevance is, in our opinion, limited because it is based on the use of a relatively simple model of the tongue, based on linear elasticity [30]. This simple account of tongue tissue and muscle mechanics is acceptable to describe in a first approximation tongue dynamics. However, it does not seem to be appropriate to simulate tongue deformations with accuracy, since tongue tissue rather behave like hyperelastic materials [31]. Another limitation of Wrench's work lies in the use of 2D views of the tongue recorded with ultrasounds. We know that ultrasounds imaging technique does not provide reliable information on flesh points, but only on the shape of the tongue contour within the beam angle, within which the tongue moves back and forth (e.g. [32]). This does not make this imaging technique a reliable tool for assessing the relationship between anatomical muscle placement and tongue deformations. From this perspective, the work of Harandi et al. [28] is more sophisticated as it used 3D MRI data and more realistic non-linear biomechanical models of the tongue and the jaw. However, in Harandi et al.'s inferred activations of the 5 parts of the Genioglossus (Fig 9 in [28]), a similarity can be observed in the medial parts of the Genioglossus (GGc and GGd) in three of the four speakers. Hence, these results do not convincingly demonstrate the necessity to consider 5 parts in the Genioglossus, and we decided to stick to the decomposition of the oblique fibers of the fan-shaped part of this muscle into 3 functional units, and we obtained satisfactory results. This does not mean that we discard the

possibility of a further refinement, but we consider that it should be based either on new physiological or anatomical evidence or on clear impossibility to properly account for measured tongue geometries.

In the context of our study, that focuses on isolated steady-state sound productions, the improved tongue geometry and the refined muscle implementation are validated by the capacity of our new model to satisfactorily account for 3D tongue shapes that were not taken into account in the design of the model, namely the two other extreme cardinal vowels (/a/ and /u/) and a front and a backstop consonant (/t/ and /k/). The approximation of the tongue shapes is pretty good in the sagittal, coronal and transversal planes. More specifically, in the mid-sagittal plane the tongue contour that is typical for each of the sounds is well accounted for, even if we observe an insufficient lowering of the tongue tip in vowel /a/. In the coronal planes, the central tongue grooving observed in the anterior part of the vocal tract for /i/, /a/ and /t/ is also well described, within the limits intrinsically due to the fact that our model is symmetrical with respect to the mid-sagittal plane, which is obviously not true for real human data. Importantly, for each of the selected sound, the constriction region of the vocal tract, i.e. the region with the smallest cross-sectional area, which is crucial for the auditory characteristics of the produced speech sound, is very well-described.

Since the selected speech sounds nicely account for the maximal range of tongue shapes reached during speech production in a French speaker, we are confident that the model is able to correctly reproduce the main characteristics of all the tongue configurations that RS achieves during speech production. Further evaluation is required though for tongue shapes achieved during swallowing, since the range of variations has been shown to be larger than in speech production [33].

## 3.2. Muscle activation patterns

We have also proposed activation patterns for the tongue and the hyoid muscles for the 5 selected elementary French sounds (/i/, /a/, /u/, /t/, /k/) (Table 4). As concerns the partitioning of the Genioglossus, we observed that the horizontal part (GGh) is activated in isolation for the production of /k/ and /u/. For these sounds, additional activation of the posterior part (GGp) would move the whole tongue body forward, preventing correct consideration of the constriction location. This result justifies the separation between the GGp and the GGh. Table 5 provides a comparison of the proposed patterns of activation for the tongue muscles with the patterns that have been proposed in the past in the literature for the production of the extreme cardinal vowels /i/, /a/ and /u/ with two other biomechanical tongue models [9,12]. Not surprisingly our patterns of activation are in good agreement with the former proposals for the muscles that are now widely known to be crucial for the articulation of these extrem cardinal vowels: Anterior Genioglossus (GGa) and Hyoglossus for vowel /a/, Styloglossus, the set made of the horizontal parts of the Genioglossus (GGh in our model which is part of the GGP in the other two models) for vowel /u/, and the posterior/horizontal (GGp/GGh) and anterior (GGa) Genioglossus for vowel /i/. These activations are also quite consistent with the EMG activations measured in American English by [24] (note that in

Table 5. **Comparison of activated muscles for cardinal vowel generation using various biomechanical models from the literature.** F: [12], B: [9], C: Our model.

|      | GGa | GGm | GGp/GGh | SG  | HG  | Vert | Trans | IL  | SL  | GH  | MH  |
|------|-----|-----|---------|-----|-----|------|-------|-----|-----|-----|-----|
| /a/  | CFB | C   | CFB     | C   | CFB | FB   | F     | CB  | -   | -   | B   |
| /u/  | FB  | -   | CFB     | CFB | F   | F    | FB    | B   | C   | C   | B   |
| /i/  | CFB | F   | CFB     | FB  | -   | F    | FB    | -   | -   | CB  | FB  |

these data the posterior Genioglossus (GGp) was not found to be active in vowel /u/ or in German by [34] (except for the anterior Genioglossus, that was not found to be active in vowel /a/). We also find in the former modeling studies an activation of the Transversalis in high vowels (/i/ and /u/), whereas activating this muscle is not necessary in our simulations. In Buchaillard et al. [9] this activation was proposed to contribute to counteract the lateral volume expansion that could occur in the *mobile tongue* in response to the compression induced by the posterior/horizontal Genioglossus in the back part of the tongue, and then to elevate the tongue blade.

Our proposed activation patterns present though a few discrepancies with the preceding modeling studies: we don't consider an activation of the Styloglossus for vowel /i/ (this is consistent with the data in [24] but not with the ones in [34]) and no activation of the anterior Genioglossus in vowel /u/ (consistent with both [24] and [34]); we propose an activation of the Superior Longitudinalis in vowel /u/ (no experimental data to compare with), and an activation of the GGm in vowel /a/, which seems to have been included in anterior Genioglossus in the studies [24] and [34]. These differences are not crucial and could be the results of some subject-specific strategies in vowel articulation.

Hence, in general, our results demonstrate the efficiency and the reliability of the *G-OBIM tongue model*, with its functional partitioning of the Genioglossus, to study motor control underlying speech articulation in a subject-specific manner in our RS. Nevertheless, we notice a number of limitations in our accounts of the articulation of the five selected speech sounds: in vowel /a/ the anterior part of the tongue is too high (this is also true, but in a lesser extent for the stop consonant /t/); in vowel /i/ and in stop consonant /k/, the tongue posture is too posterior.

## 3.3. Conclusion

We have designed a biomechanical model of the tongue in the aim to simulate with accuracy the way a specific speaker, namely our Reference Subject (RS), articulates speech sounds. This model includes a dense FE mesh made of quadratic tetrahedral elements, which faithfully reproduces the morphology of the tongue of RS. The mesh density has been determined via a convergence analysis to provide simulations with an accuracy that is comparable to one of the articulatory data collected on human speakers during speech production with Electromagnetic Articulography. While some problems do remain in the capacity of the model to account for the shape of the tongue, the model has been shown to reproduce with a generally good accuracy the speech articulations of RS. Several studies have used biomechanical models to characterize the main trends of the influence of orofacial biomechanics on general characteristics of speech production, but, to the best of our knowledge, only one study [28] has tackled this issue in a speaker-specific approach. Their approach differed from our though since they used an inversion method from several vocal tract shapes to determine tongue muscle implementation. Their approach is very accurate, but takes the risk of an over-fitting and an ad-hoc representation of muscle anatomy. Our approach did only use one tongue shape (the one of vowel /i/) to adjust a muscle implementation that was designed on the basis of anatomical and EMG findings that were repeatedly published in the literature. This muscle implementation was then evaluated on its capacity to generalize to the modeling of the articulation of four other speech sounds, which was shown to be good. Thus, we have designed a model that should be general enough to describe the production of all the speech sounds produced by our Reference Subject. This work makes possible to assess hypotheses and models of speech motor control on a specific individual, enabling us to go into details to precisely assess how speech motor control models account for well-known phenomena such as coarticulation,

variations in speaking rate or clarity, compensation for perturbations. Articulatory MRI data collected on our Reference Subject are in open-access.

## 4. Materials and methods

### 4.1. Reference subject

The model was designed to mimic the tongue of a French male speaker individual, RS, who was born in 1956. He is a scientist working on the study of speech production, with a strong expertise in articulating speech sounds in various experimental conditions. He did not report any history of motor, speech or hearing disorders.

### 4.2. Vocal tract imaging

Information about the bony and the soft tissue orofacial anatomy of RS was mainly obtained thanks to two kinds of 3D images of RS head and neck region, which were recorded while RS was in a supine position: (1) high resolution 3D MRI data ($240 \times 240 \times 157$ voxels with a spacing of 1mm in each direction), acquisition time for each phoneme: 15.328 seconds; (2) 3D Computed Tomography (CT) images (axial plans separated by 1.3 mm, with a resolution of 0.49 mm/voxel in both directions of the plan). These data were recorded in the Imaging Center of the Université Grenoble Alpes (IRMAGe), at two different times and for different purposes. An accurate representation of the geometry of the palatal vault and upper dental arch was obtained from a dental cast digitized using a high-resolution 3D scanner (SkyScan 1076 in-vivo micro-scanner). The CT images and the dental cast were collected in 2001, for the purpose of developing the biomechanical model of RS orofacial system. The MRI data were recorded in 2016, in the context of a large research project investigating the 3D shape of the vocal tract during the production of the French speech sounds. The MRI acquisition time was 15.3 seconds. The speaker was instructed to artificially sustain the phonemes for that duration (cf. [35] and [36] for more details and validation of this approach). Specifically, for all phonemes, including stops, he was instructed to repeat the utterances first a few times in a natural manner and then to freeze the consonant or the vowel in the last repetition for the duration of the MRI acquisition. The scan was launched by the operator as soon as he heard that the articulation of the phoneme of interest was being maintained. For instance, for /k/ in /i/ context, the speaker uttered [iki iki iki ik-posture hold for 15s-i].

### 4.3. Ethics statement

The CT data and the dental cast were collected in 2002 under RS informed consent. At the time, no ethical approval was required in France for this kind of data, provided that the subject was a volunteer and the physicians had agreed to carry out the acquisitions. These data have already been published in two articles in peer-reviewed journals [9,15].

The MRI data were recorded in 2016 under RS informed consent in the context of the experimental part of a pilot study, which was approved by the ethical committee of Univ. Grenoble Alpes (MammoBio MAP - VS pilot study ID RCB 2012-A00310-43, IRMaGe platform, Univ. Grenoble Alpes).

### 4.4. Geometries of the FE meshes of the main anatomical structures

A very dense and accurate FE mesh made of quadratic tetrahedral elements with a total of around 3 million nodes was designed to represent the tongue as described by the 3D MRI data for a tongue posture that was considered to be "neutral" (mandible closed and tip of the tongue slightly touching the lower incisors).

The geometries of the meshes of the bony structures (mandible, hyoid bone, and maxilla) were derived from the CT data and the dental cast. Note that these data are the same as those used by Buchaillard et al. [9].

In terms of overall topology, as compared to the last version of our tongue model [19], the new tongue mesh is more accurate and more comprehensive (see Fig 16). It provides a better definition of the sub-apical region (Fig 16, red frame), in the aim to facilitate the elevation of the tongue tip toward the palate. The description of the Hyoglossus muscle includes now a better account of its posterior triangular part (Ceratoglossus) and of its insertion on the greater horn of the hyoid bone (Fig 16, yellow frame). Finally, the orientations of the external branches of the Styloglossus were modified in order for the insertions on the skull to more accurately fit over the Stylohyoid process, as determined by the 3D description of the head provided by the CT exam (Fig 16, green frame).

Additional improvements have also been provided in the implementation of the muscles within the FE mesh, which are described later in this section. The model also provides a more realistic topological representation for the contacts of the tongue with the various other structures in the oral cavity. For the sake of completeness, the information about contacts and boundary conditions is detailed in S1 Text (Sect S3).

## 4.5. FE Mesh accuracy: Convergence analysis

When the behavior of a physical system is simulated using FE analysis, the output of the simulation is an approximation of the exact solution. To minimize the discrepancy between the simulation outcomes and the exact solution of the problem, there are important requirements on the size of the elements within the FE mesh. The smaller the elements, the more accurate the simulation. However, the larger the number of elements in the mesh, the more computationally demanding and the longer the simulations using the mesh. In order to find the best

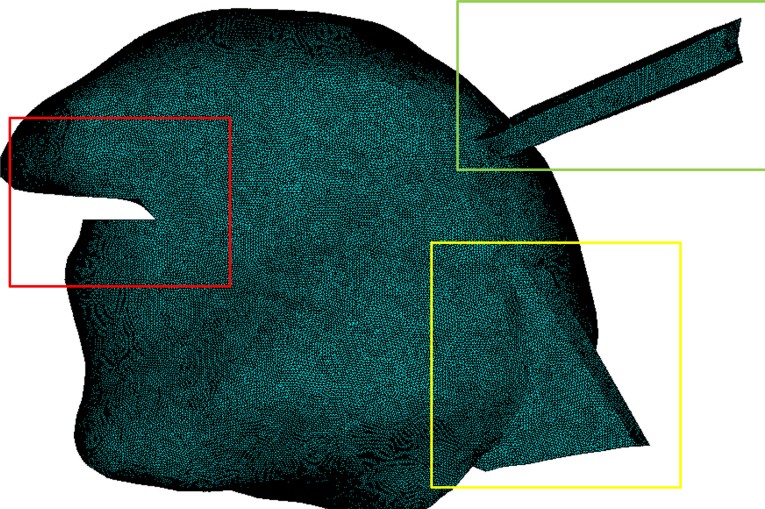

**Fig 16. Localisation of the main improvements provided to the topology of the tetrahedral tongue mesh (around 3 million nodes), as compared to the last version of the tongue model [19] in our group.** Red frame: the sub-apical region is refined. Green frame: the external branches of the Styloglossus are repositioned to refine the position of their attachment on the Stylohyoid process on the skull. Yellow frame: the posterior triangular part of the Hyoglossus (Ceratoglossus) is enlarged to insert on the greater horn of the hyoid bone.

compromise between simulation accuracy and computational costs, we have carried out a mesh convergence analysis with meshes made of quadratic tetrahedral elements. A series of simulations was conducted, in which the same constant gravity field was applied to the tongue, for a set of tongue meshes having different resolutions with a certain number of nodes in common. These meshes were designed first by strongly down-sampling the initial tongue mesh described above (3 million nodes, see Fig 16). The result was a relatively coarse mesh, which was then progressively refined, increasing its resolution by dividing the elements into a number of smaller ones. We observed at what resolution the predicted displacement magnitudes, under the effect of the gravity, of the nodes shared by all the meshes remained within an interval of variation across meshes that is smaller than 1mm. The value of 1mm was chosen because it corresponds to the extent of inaccuracy typically accepted in measuring articulatory displacements with an ElectroMagnetic Articulometer (EMA, [37]), which is generally considered a reference device for tongue movements in speech production.

Six meshes of increasing resolutions were thus generated using the *Hypermesh* software, with 7795, 57034, 187051, 436221, 1450427, and 3412064 nodes respectively. The most accurate mesh made of 3412064 nodes is considered as the reference mesh in this convergence analysis. We decided to focus on the displacements of four nodes located on the surface of the tongue, since this surface crucially determines the shape of the vocal tract and therefore the speech outcome. More specifically, nodes were selected in four regions of the vocal tract, which are phonetically relevant: apical, alveolar, velar and pharyngeal. The amplitudes of the displacements of the four nodes were measured for each of the six meshes.

## 4.6. Mechanical properties

### 4.6.1. Mechanical properties of the tongue.

Following the modeling principles proposed by Nazari et al. [38,39], each muscle is implemented in the model by defining in the mesh a specific subset of active elements, which constitutive law includes an active component in addition to the passive component of the soft tissue. In such an active element, when the muscle is recruited, a compression stress is applied in the element along a main direction, which is locally specified by a vector indicating the main direction of muscle fibers, as inferred from anatomical data. This active muscle stress is generated with a Hill-type model [40,41].

More specifically, the active element used in the current model is an improved version of the Hill-type active element proposed by Nazari et al. [20]. It makes possible for an element in which two different muscles are interwoven to apply active stress along two different directions simultaneously. Each active element is seen as a volume, in which the muscle fibers are embedded in a matrix of surrounding passive tissues. The model is implemented in the form of an ANSYS UserMat, taking into account the passive elastic properties of the tissues and the active mechanisms of force generation in one or two directions, depending on whether the element includes one or two muscles.

Thus, an element that includes a unique muscle is modeled as a transversely isotropic material with an isotropic passive behavior superimposed on an active behavior in the fiber direction. The total stress in the fiber direction is equal to the sum of the stress generated by the passive tissue and the stress due to the contractile component. It has often been suggested that the largest passive stiffness of a muscle in extension occurs along the course of the fibers [42]. However, recent uniaxial traction experiments were performed on a porcine longissimus dorsi muscle, which showed a lower passive stiffness along the direction of muscle fibers [43] than along the direction orthogonal to these fibers. Moreover, tests performed on the chicken pectoral muscle found the largest passive stiffness along a direction forming a 45° angle with the direction of the muscle fibers [44]. Similar effects could be observed in our research group

for the Genioglossus muscle of a bovine tongue, with a maximum passive stiffness in a direction close to 63° with respect to the muscle fibers' direction [45]. In the absence of repeated findings characterizing the direction of the largest passive stiffness in a muscle, we decided to take a conservative approach and to model the passive component of a muscle as an isotropic material. Based on Gerard et al.'s experimental findings [31], the same hyperelastic constitutive law was used for the passive tissue and the passive matrix of muscles, namely a Yeoh law determined by Eq 1:

$$W = C_{10}(I_1 - 3) + C_{20}(I_1 - 3)^2 + \frac{K}{2}(J - 1)^2 \tag{1}$$

where $W$ is the strain energy density, $I_1$ is the first deviatoric invariant of the right Cauchy-Green deformation tensor, $K$ is the bulk modulus and $\mathbf{J}$ is the Jacobian of the transformation (determinant of the deformation gradient tensor). The bulk modulus $K$ is given by Eq 2:

$$K = \frac{2C_{10}}{1 - 2\nu} \tag{2}$$

$\nu$ being the Poisson's ratio. Gerard et al. [31] have proposed to use for the tongue $C_{10} = 192Pa$ and $C_{20} = 90Pa$ with a Poisson's ratio $\nu = 0.49$, which assumes that soft tissues are quasi-incompressible. The Hill activation model follows the formulation of [46]. It incorporates a law of variation of the active stress as a function of the strain rate of the muscle that functionally accounts for the behavior of actin/myosin bridges in sarcomeres [47].

**4.6.2. Mechanical properties of other anatomical structures.** The other anatomical structures of the vocal tract are modeled using linear elastic laws (Table 6). For the bony structures, the moduli of elasticity are very high (values taken from [11]) so that they can be considered as almost non-deformable compared to the soft tissues.

The muscles connecting the hyoid bone to cartilages and bony structures other than the mandible ("hyoid muscles" henceforth) are also implemented in the model. They include the "infrahyoid muscles", i.e., symmetrically with respect to the mid-sagittal plane, from the center to the sides, the thyroidhyoid muscle, inserted on the upper surface of the thyroid cartilage, the sternohyoid muscle, inserted on the upper part of the sternum, and the omohyoid muscle, inserted on the scapula, and the "suprahyoid" muscles, i.e. the stylohoid muscle, inserted on the styloid process and the posterior belly of the digastric, inserted on the mastoid notch. All these muscles are modeled as 1D contractile cable elements.

The thyroid-hyoid membrane connects the upper edge of the thyroid cartilage to the hyoid bone.

The Young moduli of hyoid muscles and thyroid-hyoid membrane were defined in a functional manner to realistically account for the magnitude of the hyoid bone displacements during speech tasks. The Poisson's ratio and the density of the tissues were defined in relation to what is usually found in the literature for soft tissue (quasi-incompressibility and density equal to 1).

Table 6. Mechanical parameters of the other anatomical of the oral cavity. From [11].

| | Young Modulus $E$ (Pa) | Poisson ratio $\nu$ | Density $\rho$ ($kg/m^3$) |
|---|---|---|---|
| Hyoid bone | $1.1e10$ | 0.33 | 2000 |
| Mandible | $1.1e10$ | 0.33 | 3600 |
| Maxilla | $1.1e10$ | 0.33 | 3600 |
| Hyoid muscles | 40000 | 0.499 | 1040 |
| Thyroid-hyoid Membrane | 10000 | 0.49 | 1040 |

## 4.7. Anatomical implementation of tongue muscles

**4.7.1. General improvements.** In general, the anatomical implementation of tongue muscles in the model reproduced the choices made in our previous models (Gerard et al. [1]; Buchaillard et al. [9], Fig 1; Hermant et al. [19], Fig 5). This former implementation has been guided by anatomical data published in the literature [21,48–54] as well as by the careful analysis of the data set provided by the Visible Human Project for a female subject [55].

In addition to the above-mentioned modification of the orientation of the external branches of the Styloglossus (Fig 16), we have provided some changes that were in large part based on the careful observation of the *Acland's Video Atlas of Human Anatomy* [56]. One of these changes concerns the enlargement of the posterior part of the Hyoglossus (Ceratoglossus, described above in the yellow frame of Fig 16) and its insertion on the greater horn of the hyoid bone. Some other changes were marginal, such as the slight elevation of the insertions of the Mylohyoid on the inner part of the mandible or the refinement of the insertion of the Geniohyoid on the body of the hyoid bone. The major improvement concerns the functional partitioning of the Genioglossus and is described below.

**4.7.2. Functional partitioning of the Genioglossus.** In 1939, Abd-El-Malek [21] described this muscle as spreading "*dorsally in the substance of the tongue from root to tip, in a fan-shaped radiation both antero-posteriorly and medio-laterally*" and distinguished three subsets of fibers (anterior, intermediate and posterior). However, Abd-El-Malek did not provide explicit anatomical or histochemical foundations for this division. The Genioglossus is a bipennate muscle, in which a short quasi-horizontal tendon attached to the superior genial tubercle provides an anatomical separation between the horizontal posterior fibers and the oblique intermediate fibers [27], but the distinction between intermediate and anterior fibers remains difficult to characterize. In their morphological and histochemical study, Saigusa et al. [22] found a gradual increase of the diameter of the muscle fibers from the posterior to the anterior part of the Genioglossus. They also found that in this anterior part almost 2/3 of the fibers are fast fibers (Type II), while fast (Type II) and slow (Type I) fibers are quasi evenly represented in the posterior part. These findings support Abd-El-Malek's suggestion to not consider the Genioglossus as a whole and to distinguish different functional parts, but they do not give clear indications on how to define the separation between these parts.

As concerns functional aspects, interesting indications could be found in electromyographic studies. Miyawaki et al. [23] measured the activity of the Genioglossus during the production of various vowel-to-vowel transitions in Japanese, from 5 electrodes. Three electrodes (G1, G2, G3) were inserted to measure the Genioglossus activity over the spatial range of the "*relatively anterior fibers [...] whose attachment to the symphysis menti are relatively superior and which reach an anterior-to-medial portion of the tongue dorsum*", whereas two electrodes were inserted to cover the spatial range of the "*relatively posterior fibers*" that have "*a relatively inferior attachment and that course towards a medial-to-posterior portion of the tongue dorsum*". Interestingly, in the investigated production of vowels, electrodes G1, G2 and G3 showed very similar time patterns of activity. Another similarity was observed for electrodes G4 and G5 (Fig 3 and Fig 4 in [23]). This suggests that in speech production, functionally, two parts should be considered in the Genioglossus: a front part from the tongue tip to the medial portion of the tongue dorsum, and a posterior part from the medial portion of the tongue dorsum to the tongue root, which includes then both the intermediate and the posterior fiber bundles defined by Abd-El-Malek [21]. Interestingly, the functional coupling between the intermediate and posterior fiber bundles was also observed in a more recent electromyographic study during tongue protrusion ([57], Fig 3), a movement that is part of the production of front vowels, such as /i/ or /e/.

The first biomechanical tongue model designed in our group ([6,58]), which accounted for the 2D movements of the tongue in the mid-sagittal plane, included a description of the Genioglossus in two functional parts, which were quite similar to those considered in Miyawaki et al.'s study [23]. However, Dang & Honda [10] in their *"partial 3D"* model of the tongue demonstrated the necessity to consider at least three functional parts in the Genioglossus: an *anterior* part (GGa), which was required to lower the tongue tip without inducing any significant effect on the tongue dorsum; a *medium* part (GGm) to move the tongue dorsum and the tongue tip downward and forward; a *posterior* part (GGp) to move the pharyngeal part of the tongue forward and move the tongue dorsum and the tongue tip forward and upward. Interestingly, in Dang & Honda's model, contrary to the original suggestion by Abd-El-Malek [21], the GGp was not reduced to the horizontal fibers bundle but extended up to the back of the velar region, as was suggested by Miyawaki et al.'s results [23]. The division between the GGa and the GGm was set up within the region of the tongue that was called *"anterior fibers"* by Abd-El-Malek [21] and *"relatively anterior fibers"* by Miyawaki et al. [23]. Such a division seems to be required for a correct account of fine tongue deformations during speech production in the palatal and alveo-palatal region of the tongue, which is also called *"mobile tongue"*. This is consistent with the dominance of Type II fast muscle fibers in this region of the Genioglossus [22]. The back part of the Genioglossus is preferably used to control the posture of the whole tongue body along the front/back direction, which is also consistent with the rich proportion of Type I slow muscle fibers [22].

In the recent 3D biomechanical tongue models designed in our group [1,9,19], the functional division of the Genioglossus proposed by Dang & Honda [10] was implemented. Thanks to this approach we could show the important role of the GGa in shaping the alveo-palatal part of the tongue during the production of vowel /i/, and the antagonist roles of the GGm and the Styloglossus in the control of the elevation of the tongue dorsum in the velar region [10]. Hence, this functional division seems to be crucial to account for complex tongue shapes and movements in speech production. However, the exact locations of the boundaries between GGa, GGm and GGp are still not known.

We have used the opportunity of the current work aiming at designing a 3D biomechanical tongue model that accounts with accuracy for speech sounds' articulation by RS to improve the description of this functional partitioning of the Genioglossus. For that, we decided to rely both on anatomical observations (the separation provided by the quasi horizontal tendon between horizontal posterior and oblique intermediate fibers [27], and the difference in the proportion of slow versus fast muscle fibers between the anterior and the posterior part of the tongue [22]), and electromyographic data [23] that suggest a separation of the oblique fibers into three functional units. We thus defined a partitioning of the Genioglossus fibers into 4 functional units: the Genioglossus horizontal (GGh), corresponding to the horizontal fibers below the tendon; the Genioglossus posterior (GGp) including a large part of the posterior oblique fibers located above the tendon; the Genioglossus anterior (GGa) including the fibers inserted in the tongue tip; the Genioglossus median (GGm) located between the GGp and the GGa, covering in particular the velo-palatal part of the oblique fibers. We have used an interactive approach between simulations with the model and observations of the 3D MRI data of RS to adjust the location of the boundaries between the functional units GGp, GGm and GGa. Since the articulation of vowel /i/ is known to recruit different parts of the Genioglossus (see [10] or [12] for simulations and [24] and [23] for EMG data), the data relative to this vowel were used.

### 4.8. Evaluation of the model using 3D MRI data

To evaluate the model, we considered the 3D MRI data of /a/, /u/, /k/ in context /u/ and /t/ in context /i/, in addition to those of /i/ that are used to adjust the partitioning of the Genioglossus. The three vowels /i/, /a/, /u/ were chosen because they are universal cardinal vowels in the world languages and are associated with the most extreme tongue configurations along the front/back and the high/low directions. The stop consonants /k/ and /t/ were chosen because they involve contacts of the tongue with two different parts of the hard palate, namely the velar and the alveo-palatal parts. Tongue shapes obtained with the model for these speech sounds result from muscle activations starting from the neutral position of the tongue. These simulations were performed using transient analysis with piecewise linear muscle activation time patterns (see S1 Text, Sect S4, for details).

## Supporting information

**S1 Text. Supplementary figures and tables.**
(PDF)

## Author contributions

**Conceptualization:** Maxime Calka, Pascal Perrier, Yohan Payan.

**Data curation:** Pierre Badin.

**Funding acquisition:** Pierre Badin, Michel Rochette, Pascal Perrier, Yohan Payan.

**Investigation:** Maxime Calka, Pascal Perrier, Yohan Payan.

**Methodology:** Maxime Calka, Mohammad Ali Nazari, Pascal Perrier, Yohan Payan.

**Project administration:** Pascal Perrier, Yohan Payan.

**Resources:** Pierre Badin.

**Software:** Maxime Calka, Mohammad Ali Nazari.

**Supervision:** Michel Rochette, Pascal Perrier, Yohan Payan.

**Validation:** Maxime Calka, Pascal Perrier, Yohan Payan.

**Visualization:** Maxime Calka.

**Writing – original draft:** Maxime Calka, Pascal Perrier, Yohan Payan.

**Writing – review & editing:** Maxime Calka, Pierre Badin, Mohammad Ali Nazari, Pascal Perrier, Yohan Payan.

## Acknowledgments

We want to express special thanks to Laurent Lamalle, Research Engineer at INSERM, IRMaGe, Université Grenoble Alpes, for the acquisition of the MRI data.

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
