## [Decision Letter · Decision Letter 0]

20 Mar 2025

PCOMPBIOL-D-24-02102

The G-OBIM tongue model: an accurate open-source biomechanical model of a male human tongue

PLOS Computational Biology

Dear Dr. CALKA,

Thank you for submitting your manuscript to PLOS Computational Biology. After careful consideration, we feel that it has merit but does not fully meet PLOS Computational Biology's publication criteria as it currently stands. Therefore, we invite you to submit a revised version of the manuscript that addresses the points raised during the review process.

Please submit your revised manuscript within 30 days May 20 2025 11:59PM. If you will need more time than this to complete your revisions, please reply to this message or contact the journal office at ploscompbiol@plos.org. Please include the following items when submitting your revised manuscript:

We look forward to receiving your revised manuscript.

Kind regards,

Jonghye Woo

Guest Editor

PLOS Computational Biology

Hugues Berry

Section Editor

PLOS Computational Biology

**Additional Editor Comments:**

The reviewers provide constructive feedback, with positive overall assessments. Given their comments, a minor revision is recommended.

**Journal Requirements:**

- ® on pages: 8, 11, 14, and 15.

5) We have noticed that you have uploaded Supporting Information files, but you have not included a list of legends. Please add a full list of legends for your Supporting Information files after the references list.

6) Please ensure that the funders and grant numbers match between the Financial Disclosure field and the Funding Information tab in your submission form. Note that the funders must be provided in the same order in both places as well.

**Reviewers' comments:**

Reviewer's Responses to Questions

**Comments to the Authors:**

Reviewer #1: The tongue is responsible for major aspects of the production of all vowels, and most consonants, of the world's languages. Having accurate models of it could assist in many subfields of biology, medicine, and phonetics. Several finite-element-type models have appeared in the last several decades, but many of these, especially ones that have been made open-source are of unproven accuracy. The model presented here is impressive due to the care taken to prove convergence, and to make sure that a variety of motor actions observed experimentally with EMA are reproducible by the model. I'm also intrigued by the 4-part functional decomposition of the genioglossus muscle. I don't think enough arguments are provided for the 4 vs. 5 parts, but it's an intriguing and testable proposal. One major confusion I have is why this model is termed open-source. As far as I can tell, it's all performed in ANSYS, which is very expensive software. Is the ANSYS code what the authors plan to make available, or another implementation that could be used by a wider audience?I hope that this is clarified.

Reviewer #2: Dear Sir/Mom:

I would to inform you that this paper interesting and fairly well written.

In my opinion, the paper may be suitable for publication in the Journal of PLOS Computational Biology criteria for publication

Yours sincerely,

Dr. Waddhaah M. Alasbahy

Reviewer #3: - One of the main contributions of the paper is the partition of the Genioglossus into four functional parts. But it seems that such partitions have been proposed before. Could you please clarify what is the main difference between your approach and those mentioned in e.g., Wrench, A. A. (2024). The compartmental tongue. Journal of Speech, Language, and Hearing Research, 67(10S), 3887-3913.

In the cited paper they also proposed up to 10 divisions of the Genioglossus. Could please also comment on this?

- Just to clarify. It is mentioned that the stop consonants /t/ and /k/ where capture for ~15 seconds in context I guess something like /utu/, /iti/,…. But the durations of these consonants could be between 15-25 ms for /t/ and perhaps ~30 ms for /k/. Could you please clarify this point?

Minor comments:

- Referencing figures and abbreviations should be consistent. For instance, sometime a figure is referenced as figure 1, Fig 1., Figure 1. The abbreviation Finite Element was defined as FE, but is not used every time.

- Page 5, Line 111: The parenthesis in “…3D MRI data (240 × 240 × 157 voxels…” is not closed.

**Have the authors made all data and (if applicable) computational code underlying the findings in their manuscript fully available?**

Reviewer #1: **No: **The link provided did not work for me.

Reviewer #2: Yes

Reviewer #3: Yes

PLOS authors have the option to publish the peer review history of their article (what does this mean?). If published, this will include your full peer review and any attached files.

Reviewer #1: No

Reviewer #2: **Yes: **Waddhaah M. Al-asbahy

Reviewer #3: No

**Figure resubmission:**
---

## [Decision Letter · Decision Letter 1]

30 Jun 2025

PCOMPBIOL-D-24-02102R1

The G-OBIM tongue model: an accurate open-source biomechanical model of a male human tongue

PLOS Computational Biology

Dear Dr. CALKA,

Thank you for submitting your manuscript to PLOS Computational Biology. After careful consideration, we feel that it has merit but does not fully meet PLOS Computational Biology's publication criteria as it currently stands. Therefore, we invite you to submit a revised version of the manuscript that addresses the points raised during the review process.

Please submit your revised manuscript within 30 days Aug 30 2025 11:59PM. If you will need more time than this to complete your revisions, please reply to this message or contact the journal office at ploscompbiol@plos.org. Please include the following items when submitting your revised manuscript:

We look forward to receiving your revised manuscript.

Kind regards,

Jonghye Woo

Guest Editor

PLOS Computational Biology

Hugues Berry

Section Editor

PLOS Computational Biology

**Journal Requirements:**

1) We note that your Supplementary Figures files are duplicated on your submission. Please remove any unnecessary files from your revision, and make sure that only those relevant to the current version of the manuscript are included.

2) The file inventory includes multiple files for Figures 4 and 5. We would recommend either combining these into a single Figure 4.tiff and Figure 5.tiff  files with separate internal panels, or renumbering them as individual figures, as we are not able to publish multiple components of a single figure as separate files.

**Reviewers' comments:**

Reviewer's Responses to Questions

Reviewer #1: I am satisfied with the author's response to my concerns regarding availability, especially their promise to later provide artisynth models. I have Las gone through the respnses and changes to my other comments, and those for other reviewers, and I am satisfied with all of the responses and changes.

Reviewer #4: The paper itself is interesting and easy to follow for the most part. The new tongue should be a valuable basis for further research.

There are still some shortcomings. For the first part, please note that I am not a native speaker.

Some sections of the text are hard to understand. The following sentence [4-7] " Such a high degree of the realism will enable scientists to precisely and quantitatively assess, in a speaker-specific manner, hypotheses about speech motor control and the impact in this respect of tongue anatomy, tongue muscle arrangements and tongue dynamics." for example is grammatically incorrect ("degree of the realism") and generally hard to follow.

This section from the introduction is also challenging to understand: "The novelty of our approach lies in the fact that we want our new model to be a very faithful representation in morphological and anatomical terms of the tongue of this male individual, so that we can study precisely how this individual controls his tongue during speech production by quantitatively comparing articulatory data collected on this individual during the production of speech sequences and simulations of these same sequences carried out with the model.". While it is understandable, it is also too long and could be split up.

The abstract still introduces the goal of the paper concisely and mentions many essential aspects of the paper; some metrics and limitations could also be mentioned already.

In the introduction, there is the following passage that I already mentioned above: "The novelty of our approach lies in the fact that we want our new model to be a very faithful representation in morphological and anatomical terms of the tongue of this male individual, so that we can study precisely how this individual controls his tongue during speech production by quantitatively comparing articulatory data collected on this individual during the production of speech sequences and simulations of these same sequences carried out with the model."

I am no expert in this field. Hence, I might be wrong here, but doesn't everyone strive for "a very faithful representation"? It would be better if you would explain what you are doing to achieve a faithful representation instead of mentioning that you are trying to achieve it, in my opinion.

Regarding the results, I appreciate the thorough analysis of the mesh resolution. I also like your reasoning for selecting the mesh you ultimately chose.

I understand that optimization methods can be costly, but I am still not fond of the reliance on trial and error. The manual adjustments will be difficult to reproduce, and it is also unclear how well this generalizes to other speakers. I still appreciate that you clarified your approach a bit more later on.

The simulation for the different vowels and consonants is also fascinating. The active stress values also provide great insight. The reasoning behind the muscle activations, combined with the references, is great and easy to follow, especially for people like me who aren't experienced in this kind of work.

If I see it correctly, most assessments made in the results section are visual-based. I think that some additional quantitative results could also be helpful if creating meaningful quantitative results is feasible. This could include error margins or confidence intervals. In terms of generalizability, having a few more reference subjects would also be beneficial, but I also understand that obtaining the necessary data can be very challenging.

Lastly, for me, it's not clear why you chose those particular vowels and consonants for your analysis. Edit: This is answered later on. However, I still think that this could be directly mentioned in the results section.

The discussion and the conclusion provide great additional insight into why you decided to subdivide the structure into four parts instead of five. The reasoning is sound, and your results also prove that your approach is good.

Section four offers some great additional insights. In my opinion, it is also a bit easier to read compared with some of the other sections.

Generally, I think that the paper only needs some rewording and, if possible, a few more quantifiable results if there are concrete things that can be easily quantified in the results section. The visual results are great but don't offer enough insight, in my opinion.

Reviewer #5: N/A

**Have the authors made all data and (if applicable) computational code underlying the findings in their manuscript fully available?**

Reviewer #1: Yes

Reviewer #4: Yes

Reviewer #5: None

PLOS authors have the option to publish the peer review history of their article (what does this mean?). If published, this will include your full peer review and any attached files.

Reviewer #1: **Yes: **Khalil Iskarous

Reviewer #4: No

Reviewer #5: No

**Figure resubmission:**
---

## [Editor Report · Decision Letter 2]

30 Jul 2025

Dear Mr. CALKA,

We are pleased to inform you that your manuscript 'The G-OBIM tongue model: an accurate open-source biomechanical model of a male human tongue' has been provisionally accepted for publication in PLOS Computational Biology.

Best regards,

Jonghye Woo

Guest Editor

PLOS Computational Biology

Hugues Berry

Section Editor

PLOS Computational Biology

---

## [Editor Report · Acceptance letter]

PCOMPBIOL-D-24-02102R2

The G-OBIM tongue model: an accurate open-source biomechanical model of a male human tongue

Dear Dr Calka,

I am pleased to inform you that your manuscript has been formally accepted for publication in PLOS Computational Biology. Your manuscript is now with our production department and you will be notified of the publication date in due course.

With kind regards,

Anita Estes
